# Cocaine-Induced Midline Destructive Lesions (CIMDL): A Real Challenge in Diagnosis

**DOI:** 10.3390/ijerph18157831

**Published:** 2021-07-23

**Authors:** Michele Di Cosola, Mariateresa Ambrosino, Luisa Limongelli, Gianfranco Favia, Andrea Santarelli, Roberto Cortelazzi, Lorenzo Lo Muzio

**Affiliations:** 1Department of Clinical and Experimental Medicine, University of Foggia, 71122 Foggia, Italy; michele.dicosola@unifg.it (M.D.C.); mariateresaambrosino@libero.it (M.A.); 2Department of Interdisciplinary Medicine, Odontostomatology Unit, University of Bari “Aldo Moro”, 70121 Bari, Italy; luisannalimongelli@gmail.com (L.L.); gianfranco.favia@uniba.it (G.F.); roberto.cortelazzi@uniba.it (R.C.); 3Department of Clinic Specialistic and Stomatological Sciences, Polytechnic University of Marche, 60121 Ancona, Italy; andrea.santarelli@univpm.it

**Keywords:** cocaine, palate perforation, cocaine-induced midline destructive lesion, CIMDL

## Abstract

The prolonged use of intranasal cocaine can destroy the nasal architecture with the erosion of the palate, turbinates, and ethmoid sinuses causing cocaine-induced midline lesions (CIMDL). The CIMDL display a clinical pattern mimicking variable diseases. The aim of this study was to highlight the difficulties in reaching a correct diagnosis through the evaluation of eight new cases. The diagnostic procedures followed in these patients included: detailed medical history, clinical and histological examination, computed tomography and magnetic resonance imaging, laboratory findings (complete blood count, sedimentation rate, antinuclear antibody test, rheumatoid factor, venereal disease research laboratory test, leishmaniasis and fungal serology, antineutrophil cytoplasmic antibodies ANCA test), and chest X-ray. All patients complained of epistaxis, halitosis, nasal scabs and obstruction, decreased sense of smell and/or taste, oro-nasal regurgitation of solids and liquids with recurrent sinus infections, and chronic facial pain. On clinical examination, all patients showed palate perforation with variable nasal structure involvement and presented a strong positivity for ANCA tests with a p-ANCA pattern. The followed protocol for the CIMDL diagnosis allowed for a relatively quick and conclusive diagnosis in all patients. A multidisciplinary approach is mandatory in the management of CIMDL, involving dental professionals, maxillofacial surgeons, and psychologists.

## 1. Introduction

Cocaine is one of the most used drugs. In fact, its market, according to data released by the World Drug Report, had an important increase, reaching an all-time high in 2017 with a production of 1976 tons. The increase compared to 2016 was 25%. The regular use of cocaine can have serious orofacial effects, such as the perforation of the nasal septum and palate, gingival lesions and tooth surface erosion, in addition to being associated with changes in the sense of smell and chronic sinusitis.

Intranasal inhalation, or “snorting”, is the most frequently used route of administration of powdered cocaine. Consequently, the adverse effects of cocaine on the nasal tract are common [1] due to the intense local vasoconstrictive effects and the mucosal irritation. Even as early as 3 weeks after the start of regular cocaine “snorting”, it may lead to inflammation and the ulceration of the nasal mucosa [2], whereas prolonged cocaine inhalation may cause ischemic osseocartilaginous necrosis with the subsequent perforation of midline structures [3]. These lesions, described for the first time in 1988, are called cocaine-induced midline destructive lesions (CIMDL) [4]. 

The pathogenesis of CIMDL is intriguing and multifactorial. Vasoconstriction, chemical irritation, mechanical trauma from high velocity inhalation, toxic effect of adulterants mixed with cocaine (amphetamines or caffeine), secondary bacterial infection, antineutrophil cytoplasmic antibodies (ANCA) formation, immunosuppression, and osteoblast inhibition all appear to play prominent roles in inducing destruction [5]. Cocaine, similar to many local anaesthetics, interacts with voltage-sensitive Na channels and activates an intrinsic apoptotic pathway, which occurs when the drug is in contact with the respiratory epithelial cells of abusers. The apoptosis-inducing effects of cocaine are dose and time dependent. In summary, CIMDL seem to be the result of a necrotising inflammatory tissue response triggered by cocaine abuse in a subset of patients predisposed to producing ANCA [6]. 

The clinical presentation ranges from intranasal crusting, foul exudate, epistaxis, nasal scabs and obstruction, chronic sinusitis to headache, fever, saddle nose deformities, nasocutaneous fistulas, necrotizing ulcerative lesions, and septal perforation. In more severe cases, the destruction extends to the middle and superior turbinates, the lateral wall of the nose, and the hard palate, causing dysphagia and nasal reflux [7]. Bone may be exposed, teeth are often missing, or severely decayed, and periodontal disease is relatively common due to oral neglect [8]. Common radiologic findings include opacification of the paranasal sinuses and mucoperiosteal thickening of the nasal and paranasal inflamed membranes [5]. Magnetic resonance imaging (MRI) can detect areas of an abnormal nasal or paranasal mucosa, and, in some patients, progressive “centrifugal” involvement of the lateral nasal walls and floor with eventual erosion of the hard and soft palate [9]. Soft tissue biopsies consistently reveal necrosis, chronic inflammation, granulation tissue, ulceration, and occasionally dense fibrosis [5].

According to Smith, besides the toxicological confirmation of cocaine abuse, the diagnosis of CIMDL is based on the clinical and radiological detection of at least two of the following findings: palatal destruction, lateral nasal wall, choanae and paranasal sinus necrosis, resulting in the collapse of the nose due to the absence of nasal septum perforation, oronasal fistulas, and even complete midface necrosis [4]. 

These CIMDL display a clinical pattern, mimicking aggressive neoplastic and necrotizing inflammatory diseases with positive ANCA tests, radiographic abnormalities, chiefly in the nasal septum and turbinates, or marked histopathological changes, such as Wegener’s granulomatosis (WG), atrophic rhinitis, tertiary syphilis, systemic lupus erythematous, lymphoma, tuberculosis, leishmaniasis and fungal infections (aspergillosis, mucormycosis, blastomycosis) [7,10,11,12]. ANCA are detectable in most patients with WG as well as in most patients with CIMDL. Whereas a cytoplasmic (cANCA) staining pattern predominates in WG, a perinuclear (pANCA) staining pattern is typical in CIMDL. ANCA, specifically for neutrophil elastase (NE), represent a valuable diagnostic marker for CIMDL and are almost never detectable in patients with WG [13]. CIMDL are often overlooked in clinical practise, particularly if a history of cocaine inhalation is not sought or volunteered. Although several reconstructive procedures or maxillary obturator prosthesis have been proposed to repair palatal defects, the best therapeutic choice is still a matter of discussion [10,14,15,16,17,18,19,20].

In this study, eight patients have been evaluated in order to investigate the correlation among clinical, histological, laboratory, and radiological findings, highlighting the difficulties in reaching diagnosis and differential diagnosis in patients with CIMDL.

## 2. Materials and Methods

A retrospective study on clinical records of patients diagnosed with CIMDL between 2001 and 2014 was conducted at the Oral medicine units of three Italian Universities: Ancona, Bari, and Foggia. Patients with CIMDL diagnosis confirmed by clinical, histological, laboratory, and radiological findings were enrolled in the study. 

The diagnostic procedures followed in these patients with midline destructive lesions included detailed medical history, extra- and intra-oral examination, histological examination on mucosal biopsies, computed tomography (CT) and MRI, laboratory findings (complete blood count, sedimentation rate, antinuclear antibody (ANA) test, rheumatoid factor, venereal disease research laboratory (VDRL) test, leishmaniasis and fungal serology, ANCA test), and chest X-ray.

Information retrieved from clinical records included: gender, age, personal medical history, form of cocaine assumption, duration of substance abuse, daily dose, symptoms, location and extension of the lesions, clinical and radiological outcomes.

## 3. Results

In this study eight patients with CIMDL were identified. Their characteristics are shown below and summarized in Table 1.

Five patients were males and three females and the age ranged from 25 to 46 years. Initially, just five patients admitted to cocaine use. All patients had a long history of cocaine abuse by inhalation: from 6 to 20 years, with a daily dose varying from 0.5 (in the early period) to 6 grams. All patient complained of epistaxis, halitosis, nasal scabs and obstruction, decreased sense of smell and/or taste, oro-nasal regurgitation of solids and liquids with recurrent sinus infections, and chronic facial pain. On clinical examination all patients showed palate perforation with variable nasal structures involvement (Figure 1A,B).

In two cases, the chronic cocaine inhalation caused a complete erosion of hard palate extended until the soft palate, with destruction of bony and cartilaginous septum, ethmoidal cells, turbinates and medial maxillary sinus walls, revealed by CT scan (Figure 2a). The nasal involvement resulted in typical saddle-nose deformities with columellar retraction and nasal collapse (Figure 2b).

Tests for ANA, rheumatoid factor, and VDRL were all negative. Leishmaniasis and fungal serology were negative, too. All patients presented a strong positivity for ANCA tests (through indirect immunofluorescence) with a p-ANCA pattern, whereas chest radiographies were normal. Biopsies of nasal mucosal lesions revealed necrotic tissue with an inflammatory infiltrate, without granuloma or vasculitis in all patients. 

A psychologist investigated the real intention of these patients to sustain a long period of abstinence after the surgery.

During the mean follow-up period of 32 months after the surgical intervention, only one patient showed a relapse of the lesions because he continued a regular cocaine use throughout this illness, whereas the other seven patients did not present any progression of the lesions. 

## 4. Discussion

Cocaine is one of the most addictive and dangerous illicit drugs currently used. The European Commission of drug abuse stated that 13 million adults have used cocaine at least once in their lifetime. Systemically, it can cause acute myocardial infarction, cerebral vascular accident, cardiac arrhythmias, and other diseases [9]. The most frequently used route of administration of cocaine is intranasal inhalation, consequently, the adverse effects on the nasal tract are common [1]. Locally, the prolonged use of intranasal cocaine can destroy the nasal architecture with erosion of the palate, turbinates, and ethmoid sinuses (CIMDL) [9]. Despite the widespread cocaine abuse by inhalation, reports on lesions involving the nasal tract are rare and the incidence of CIMDL remains unknown. Many addicts do not look for consultation because they have minimal asymptomatic lesions, because they have found their own methods for managing their complications, because seeking medical treatment is equivalent to accepting the drug problem, or because they fear that doctors will force them to stop using the drug, or that the hospital will alert legal authorities of the illicit habit. Many who do seek medical attention, perhaps due to the severity of their condition, never reveal their drug history [2] and ultimately receive treatment for some other erroneous diagnosis [9]. 

Evaluation of these patients should be multidisciplinary, the essential element for a correct diagnosis of CIMDL would be clinical history and exhaustive clinical examination. However, it is very unlikely that cocaine abusers admit their dependence, or volunteer information on the duration of abuse and dose, because of their poor compliance with physicians. CIMDL generate great diagnostic difficulties because clinical aspect, radiological findings, histopathological results, and laboratory analysis do not give conclusive information. Clinically, CIMDL may mimic aggressive neoplastic and necrotizing inflammatory disease such as WG, atrophic rhinitis, tertiary syphilis, systemic lupus erythematous, lymphoma, tuberculosis, leishmaniasis, deep fungal infections, and others. The correct diagnosis can be reached by a detailed examination of the clinical, radiological, analytical, and histopathological findings [11]. All patients suspected of CIMDL should be evaluated following a diagnostic protocol including:Detailed medical history to find out the cocaine inhalation abuse;Accurate extra- and intra-oral examination to evaluate the tissue damages;Histological examination on mucosal biopsies to keep out diseases with typical features;CT and MRI examination;Laboratory findings:◦Complete blood count and sedimentation rate;◦ANA test to exclude systemic lupus erythematous;◦VDRL test to exclude tertiary syphilis;◦Leishmaniasis and fungal serology;◦ANCA test to exclude WG;◦Chest X-ray to exclude tuberculosis or WG.


This protocol allowed a relatively quick and conclusive diagnosis in all patients, even if the adequate medical history is often necessary for the exact diagnosis. In fact, differential diagnosis between oncologic or infectious disorders and CIMDL is easy for the characteristic histopathological and microbiological features, while there are several problems with the differentiation between CIMDL and GPA. ANCA positivity is specific for granulomatosis with polyangiitis (GPA) and Wegener (WG) [21,22]. However, Wiesner et al. showed that 84% the CIMDL patients were positive for ANCA test [13]. A distinct p-ANCA pattern targeting elastase (HNE ANCA) is often present in CIMDL [13,23]. These data could complicate the differential diagnosis, but the histopathological aspect can help us in the differential diagnosis. CIDML patients present extensive necrosis, while acute inflammatory and chronic perivascular infiltrate are rare, such as other GPA-associated features (granulomas, giant multinucleated cells or fibrinoid changes).

It is very important that the psychologist support the patient during the cocaine detox period. In fact, the absence of lesion progression in seven patients was due to their good compliance in stopping cocaine use, allowing for the function rehabilitation of the defects through surgery reconstruction or maxillary obturator prosthesis, while the continuation of cocaine consumption in the eighth patient determined the persistence of perforation. 

## 5. Conclusions

CIMDL is an emerging health problem due to cocaine abuse. The diagnosis of these lesions can be challenging because numerous conditions can present with similar signs and symptoms. In fact, several diseases, such as granulomatosis with polyangiitis (Wegener’s), sarcoidosis, extranodal NK/T-cell lymphoma (ENKTL), infectious bacterial diseases (syphilis, leprosy, rhinoscleroma, tuberculosis, actinomycosis), infectious fungal diseases (histoplasmosis, mucomycosis, blastomycosis, coccidiomycosis) and infectious parasitic diseases (leishmaniasis, myiasis) can determine the perforation of the mid-line palate. Differential diagnosis between oncologic or infectious disorders and CIMDL is easy for the characteristic histopathological and microbiological features. The use of specific tests, such as HNA ANCA, can facilitate the differential diagnosis of CIMDL with GPA or WG, together with the same histopathologic aspect of the specimens. 

A multidisciplinary approach is mandatory in the diagnosis and subsequent management of CIMDL, involving dental professionals for the early diagnosis of palatal perforations and for the prosthetic rehabilitation, maxillofacial surgeons for the surgical reconstruction, and psychologists to investigate the real intention of patients to obtain the independence from cocaine.

## Figures and Tables

**Figure 1 ijerph-18-07831-f001:**
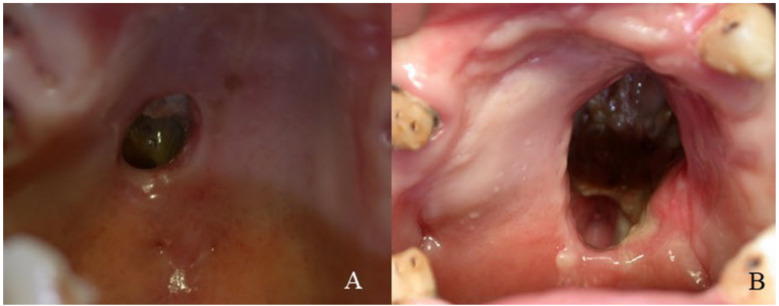
(**A**) Clinical aspect of case No. 2; (**B**) clinical aspect of case No. 4.

**Figure 2 ijerph-18-07831-f002:**
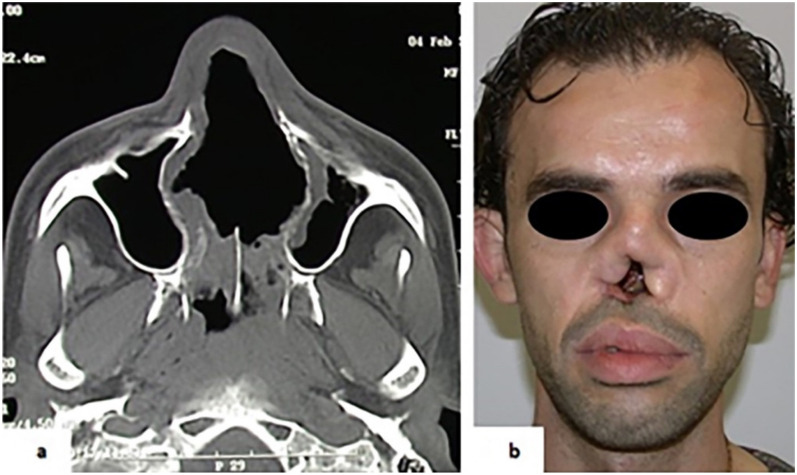
(**a**) Case 5. Destruction of bony and cartilaginous septum revealed by CT scan. (**b**) Typical saddle-nose deformities with columellar retraction and nasal collapse.

**Table 1 ijerph-18-07831-t001:** Patients’ clinical data.

Patients characteristics	Data	%
Males	5	62.5%
Females	3	37.5%
Males-to-females ratio	1.67:1	
Age range	25–46 years	
Mean age	36.37 ± 7.13 SD	
Mean duration of cocaine abuse	13 years	
Mean daily dose	3.5 grams	

SD: standard deviation.

## Data Availability

Restrictions apply to the availability of these data. Data are available from the corresponding author.

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
