# Peer review of "Cocaine-Induced Midline Destructive Lesions (CIMDL): A Real Challenge in Diagnosis"

_ijerph, 2021, doi:10.3390/ijerph18157831_

Round 1
Reviewer 1 Report
This study aimed to investigate the correlation among clinical, histological, laboratory, and radiological findings, highlighting the difficulties in reaching diagnosis and differential diagnosis in patients with CIMDL.
Title: according to the main objective of this study, the subject is the diagnosis of CIMDL, so I recommend to improve the title by informing the reader about the diagnosis.
Abstract: The objective is missing from the abstract.
Table 1: Please, insert the legend explaining the abbreviations: N, SD.
Discussion: Avoid ‘etc’ terms.
The discussion of this study presents a poor correlation the results found and the scientific literature. Consider adding more arguments discussing each test aimed at the study.
Conclusion: The conclusion is the part of the paper that answer the main question (objective) of the study, so the conclusion presented is focused on the treatment of CIMDL, which was not investigated and discussed. The objective is related to the diagnosis.
Author Response
Title: according to the main objective of this study, the subject is the diagnosis of CIMDL, so I recommend to improve the title by informing the reader about the diagnosis.
Reply: we modified the title ‘’Cocaine-Induced Midline Destructive Lesions (CIMDL): report of eight cases’’ in ‘’ Cocaine-Induced Midline Destructive Lesions (CIMDL): a real challenge in diagnosis’’
Abstract: The objective is missing from the abstract.
Reply: we reported the aim this study in the abstract section.
Table 1: Please, insert the legend explaining the abbreviations: N, SD.
Reply: We substituted ‘’N’’ with ‘’Data’’, and added the ‘’*SD: standard deviation’’ in Table 1.
Discussion:
Avoid ‘etc’ terms.
Reply: We substituted the term ‘’etc.’’ with ‘’and other diseases’’
The discussion of this study presents a poor correlation the results found and the scientific literature. Consider adding more arguments discussing each test aimed at the study.
Reply: We improved the Discussion section adding arguments about differential diagnosis.
Conclusion: The conclusion is the part of the paper that answer the main question (objective) of the study, so the conclusion presented is focused on the treatment of CIMDL, which was not investigated and discussed. The objective is related to the diagnosis.
Reply: We added ‘’CIMDL is an emerging health problem due to cocaine abuse. The diagnosis of these lesions can be challenging because numerous conditions can present with similar signs and symptoms. In fact several disease such as granulomatosis with polyangiitis (Wegener’s), sarcoidosis, extranodal NK/T-cell lymphoma (ENKTL), infectious bacterial diseases (syphilis, leprosy, rhinoscleroma, tuberculosis, actinomycosis), infectious fungal diseases (histoplasmosis, mucomycosis, blastomycosis, coccidiomycosis) and infectious parasitic diseases (leishmaniasis, myiasis) can determine perforation of the mid-line palate. Differential diagnosis between oncologic or infectious disorders and CIMDL is easy for the characteristic histopathological and microbiological features. The use of specific tests such as HNA ANCA can facilitate the differential diagnosis of CIMDL with GPA or WG, together to same histopathologic aspect of the specimens.’’.
Reviewer 2 Report
In this case report, the authors describe eight cases of cocaine-induced midline destructive lesions (CIMDL), and explain a protocol for its diagnosis, which involves a multidisciplinary approach.
General comments:
This paper addresses an important topic, as there is great interest in improving the precision and speed of the diagnosis of CIMDL. As well pointed out by the authors, a multidisciplinary approach is paramount.
Specific comments:
- I would suggest more description in the text, as well as more figures in the result section, that represent the correlation between the different diagnosis methods, and explore the difficulties.
- In the Introduction section, I would suggest adding more information, especially more identification in the literature, to better contextualize the study and its relevance.
- As a suggestion, in the Discussion section, the authors could better explore the difficulties of correctly diagnosing CIMDL, including the psychological aspect and differential diagnosis, contextualizing it in the current literature.
- Overall, this paper has a potential to be considered for publication due to the importance of improvement in the diagnosis of CIMDL.
Author Response
Specific comments:
1. I would suggest more description in the text, as well as more figures in the result section, that represent the correlation between the different diagnosis methods, and explore the difficulties.
Reply: We reported in Discussion section problematics concerning differential diagnosis of CIMDL.
2. In the Introduction section, I would suggest adding more information, especially more identification in the literature, to better contextualize the study and its relevance.
Reply: We added in the Introduction section information about the importance of this subject.
3. As a suggestion, in the Discussion section, the authors could better explore the difficulties of correctly diagnosing CIMDL, including the psychological aspect and differential diagnosis, contextualizing it in the current literature.
Reply: We improved the Discussion section adding information about psychological aspect, differential diagnosis of CIMDL.
Round 2
Reviewer 2 Report
The authors have considered the suggestions and made adequate changes, improving the overall quality of this manuscript.